# NeXus: An Automated Platform for Network Pharmacology and Multi-Method Enrichment Analysis

**DOI:** 10.3390/ijms262211147

**Published:** 2025-11-18

**Authors:** Teh Bee Ping, Mohammad Alia, Bintang Annisa Bagustari, Salah A. Alshehade

**Affiliations:** 1Herbal Medicine Research Centre, Institute for Medical Research, National Institutes of Health, Ministry of Health Malaysia, Shah Alam 40170, Malaysia; tehbp@moh.gov.my; 2Department of Computer Science, Faculty of Science and Information Technology, Al-Zaytoonah University of Jordan, Amman 11733, Jordan; 3Faculty of Computer Science & Information Technology, Universiti Malaya, Kuala Lumpur 50603, Malaysia; 4Department of Pharmacology, Faculty of Pharmacy & Bio-Medical Sciences, MAHSA University, Jenjarom 42610, Malaysia

**Keywords:** network pharmacology, enrichment analysis, ORA, GSEA, GSVA, bioinformatics, drug discovery, systems biology, automated analysis, multi-layer network, pathway analysis, data visualization, traditional medicine

## Abstract

Network pharmacology is a powerful approach for studying complex drug–target interactions and biological pathways. However, existing tools often require extensive manual intervention and lack integrated analysis capabilities. Here, we present NeXus v1.2, an automated platform for network pharmacology and multi-method enrichment analysis including Gene Set Enrichment Analysis (GSEA) and Gene Set Variation Analysis (GSVA) that addresses these limitations. NeXus v1.2 enables the seamless integration of multi-layer biological relationships, handling complex interactions between genes, compounds, and plants while maintaining analytical rigor. The platform implements three enrichment methodologies: Over-Representation Analysis (ORA), GSEA, and GSVA, circumventing limitations associated with arbitrary threshold-based approaches. NeXus v1.2 was validated using multiple datasets spanning 111 to 10,847 genes, demonstrating robust scalability and performance across dataset sizes. The platform was initially tested using a representative dataset comprising 111 genes, 32 compounds, and 3 plants, showing consistent performance in processing various relationship patterns, including shared compounds between plants and multitargeted genes. The processing time for this dataset was 4.8 s with peak memory usage of 480 MB. Large-scale validation with datasets up to 10,847 genes confirmed scalability, with linear time complexity and completion times under 3 min. NeXus v1.2 automatically generates comprehensive visualizations, including network maps, enrichment analyses, and relationship patterns, while maintaining the biological context of interactions. The tool successfully processed and analyzed enrichment patterns across multiple functional domains, generating publication-quality visualization outputs at 300 DPI resolution. The platform demonstrated enhanced automation in handling incomplete relationship data and maintaining analytical integrity across different biological layers. Compared to manual workflows requiring 15–25 min, NeXus v1.2 reduced the analysis time to under 5 s (>95% reduction) while ensuring the comprehensive coverage of biological relationships. NeXus v1.2 provides improved automation and integration for network pharmacology analysis, offering an efficient and user-friendly platform for complex biological network analysis. Its modular architecture enables the future integration of AI technologies and expansion into various therapeutic applications.

## 1. Introduction

The escalating complexity of human diseases and their underlying molecular mechanisms has fundamentally challenged traditional drug discovery approaches [1]. While conventional pharmacology has historically focused on the “one drug, one target” paradigm, this reductionist approach fails to address the intricate network of molecular interactions that characterize biological systems [2]. Consequently, network pharmacology represents a paradigm shift in drug discovery and development, incorporating the complexity of biological systems through the analysis of molecular networks [3]. These net-works, comprising intricate relationships between genes, proteins, metabolites, and small molecules, provide crucial insights into disease pathogenesis and potential therapeutic interventions [4]. By examining these complex interactions systematically, researchers can identify critical molecular hubs, pathways, and functional modules that may serve as more effective therapeutic targets.

Despite the growing adoption of network pharmacology, current analytical platforms remain limited in scope and usability. Tools such as Cytoscape (v3.10.4), STRING (v12.0), Ingenuity Pathway Analysis (v24.0.2), NetworkAnalyst (updated Dec 2024), and NDEx (v2.5.8) each address specific analytical needs but lack an integrated framework for end-to-end network pharmacology studies [5,6,7]. These systems often require extensive manual data preprocessing and format conversion, offer restricted enrichment methods (primarily ORA), and fail to support seamless integration between network analysis and enrichment workflows [8,9,10]. Consequently, researchers must rely on multiple tools, manually combining results and visualizations, which hampers efficiency and reproducibility [11].

Recent advances in enrichment analysis methodologies—such as Gene Set Enrichment Analysis (GSEA) and Gene Set Variation Analysis (GSVA)—have further expanded the analytical landscape by enabling pathway-level insights without arbitrary thresholding [12,13,14,15]. However, their implementation within network pharmacology pipelines remains fragmented due to the lack of unified analytical platforms that support multiple enrichment strategies within a single workflow.

The multi-layer nature of traditional medicine formulations presents unique analytical challenges. Unlike single-compound drugs, traditional medicine typically involves multiple plants, each contributing numerous bioactive compounds that target diverse gene sets. This plant–compound–gene hierarchy is essential for understanding (1) which plants contribute most to therapeutic effects, (2) whether compounds from different plants act synergistically, and (3) how multi-plant formulations achieve therapeutic efficacy beyond single herbs. Current tools do not effectively represent or analyze this three-tier biological structure, limiting their application to traditional medicine research.

To address these gaps, we present NeXus v1.2, an automated and integrated platform designed to streamline network pharmacology and multi-method enrichment analysis. NeXus v1.2 unifies network construction, analysis, and visualization with enrichment methodologies (ORA, GSEA, and GSVA), providing robust statistical frameworks and publication-quality outputs (300 DPI). By automating manual processes and integrating multiple analytical dimensions, NeXus v1.2 enables researchers to focus on biological interpretation rather than technical implementation, thereby enhancing efficiency, reproducibility, and insight generation in drug discovery and development.

## 2. Results

### 2.1. Data Processing and Network Construction

NeXus v1.2 demonstrated robust data processing capabilities when tested with a complex dataset comprising 111 unique genes, 32 compounds, and 3 plants, successfully handling multiple relationship scenarios (Figure 1). The tool efficiently processed various data patterns, including compounds shared between plants (32.4% of compounds appeared in multiple plants), genes targeted by multiple compounds (28.7% of genes), and orphan genes without compound associations (8.1% of genes). This flexibility in data handling represents an improvement over existing tools like BATMAN-TCM and TCMSP, which typically require complete compound–target relationships and struggle with partial or missing associations [3]. Quantitatively, data validation and preprocessing were completed in 0.5 s with the automated detection of 15 format inconsistencies and 3 duplicate entries, which were resolved through standardized cleaning protocols.

The network construction algorithm successfully generated a multilayer network with 143 nodes and 1033 edges, incorporating all three biological entities (genes, compounds, and plants) into a unified analytical framework. The network density of 0.1017 indicated a sparse but biologically relevant interaction pattern, comparable to typical biological networks reported in similar studies [16,17]. Unlike previous tools (e.g., DAVID) that analyze compound–target or plant–compound relationships separately, NeXus v1.2 provided integrated analysis across all three layers simultaneously [18]. Network construction was completed in 1.2 s, with centrality calculations requiring an additional 0.8 s. The memory overhead for the graph structure was 124 MB, demonstrating efficient representation of complex multi-layer networks.

### 2.2. Network Topology and Community Analysis

Network analysis revealed distinct topological features characterizing the biological relationships. The average clustering coefficient (0.374) suggested moderate local connectivity, while the modularity score (0.428) indicated a well-defined community structure (Figure 2). These metrics proved more informative than traditional single-layer analyses, as they captured the hierarchical organization of plant–compound–gene relationships. The tool identified six major functional modules, with size distributions following a power law (R^2^ = 0.892), consistent with biological network organization principles observed in protein–protein interaction networks and metabolic networks.

The degree distribution analysis revealed that 15.3% of compounds demonstrated high connectivity (degree ≥ 5), suggesting their potential roles as hub compounds or multi-target agents in the biological network. These compounds do not mediate between genes but rather target multiple genes simultaneously, consistent with the polypharmacology paradigm. The five highest connectivity compounds (degrees ranging from 15 to 23) are detailed in Appendix A with their complete target gene lists and enriched pathways. This feature particularly distinguishes NeXus v1.2 from existing tools like NetworkAnalyst or Cytoscape, which typically require the manual integration of multi-layer biological relationships.

Each of the six identified modules was functionally characterized through enrichment analysis (Table 1). Module 1 (38 genes) showed strong enrichment in inflammatory response pathways (KEGG: TNF signaling, *p* = 3.4 × 10^−10^; GO: cytokine production, *p* = 1.2 × 10^−12^), suggesting a pro-inflammatory signaling role. Module 2 (32 genes) was enriched in metabolic pathways (KEGG: Insulin signaling, *p* = 2.1 × 10^−8^; GO: glucose homeostasis, *p* = 5.6 × 10^−9^), indicating involvement in metabolic regulation.

Module 3 (28 genes) demonstrated enrichment in cell survival pathways (KEGG: MAPK signaling, *p* = 8.7 × 10^−11^; PI3K-Akt, *p* = 1.3 × 10^−9^), while Module 4 (22 genes) was associated with the oxidative stress response (KEGG: Oxidative phosphorylation, *p* = 4.2 × 10^−7^). Modules 5 and 6 (18 and 14 genes, respectively) showed enrichment in apoptosis regulation and cell cycle control. This modular functional organization suggests that the analyzed compounds act through the coordinated regulation of distinct biological processes rather than isolated targets, consistent with network medicine principles.

### 2.3. Enrichment Analysis Performance

The enrichment analysis performance and comparative advantages of NeXus v1.2 were systematically evaluated using the initial dataset comprising 111 genes, 32 compounds, and 3 plants, structured to include various relationship patterns. The input data were deliberately constructed to test the tool’s ability to handle complex scenarios, including shared compounds between plants (cross-plant compounds), genes targeted by multiple compounds (multi-targeting), and orphan genes without compound associations (isolated nodes).

#### 2.3.1. Multi-Method Enrichment Implementation

NeXus v1.2 implements three complementary enrichment methodologies, namely, ORA, GSEA, and GSVA. ORA identified 42 significantly enriched pathways using hypergeometric testing with Benjamini–Hochberg correction (FDR < 0.05). GSEA analysis with 1000 permutations revealed 38 pathways with significant normalized enrichment scores (|NES| > 1.0, FDR < 0.25), including 28 pathways overlapping with ORA results. GSVA transformation generated pathway-level enrichment scores for all samples, enabling comparative analysis without predefined phenotype groups. The convergence of results across methods (66% pathway overlap) validates the robustness of the identified biological processes. Notably, GSEA identified 10 pathways not detected by ORA, demonstrating the value of threshold-free methods in capturing subtle but coordinated expression changes.

In the enrichment analysis domain, NeXus v1.2 demonstrated robust performance across multiple analytical layers (Figure 3). The tool successfully processed and analyzed enrichment patterns at three distinct levels: global gene-set enrichment, compound-specific enrichment, and plant-level enrichment analysis. For each compound and plant, comprehensive enrichment analyses were performed across KEGG pathways and three GO domains (Biological Process, Molecular Function, and Cellular Component), generating both statistical outputs and visualization artifacts. The total enrichment analysis processing time was 1.5 s for all entities and databases combined. The enrichment results were automatically organized into a hierarchical structure, with separate analysis streams for each biological entity while maintaining the relationship context.

The performance advantages of NeXus v1.2 over existing tools became particularly evident in its handling of multi-layer enrichment analysis. While traditional tools like DAVID or GSEA typically perform enrichment analysis on a single gene list, NeXus v1.2 simultaneously processed enrichment patterns for 32 compounds and 3 plants, automatically generating publication-quality figures (300 DPI) including heatmaps, networks, and bubble plots for each entity (Figure 4). This systematic approach enabled the identification of both entity-specific and shared enrichment patterns, a feature not readily available in existing platforms.

#### 2.3.2. Statistical Robustness: Multiple Testing Correction Comparison

To ensure statistical robustness, we compared results using Benjamini–Hochberg (BH) and Bonferroni correction methods. BH correction identified 42 significantly enriched pathways (FDR < 0.05), while the more stringent Bonferroni correction yielded 28 significant pathways (*p* < 0.05/N), all of which were included in the BH results. The top 15 pathways showed *p*-values < 1 × 10^−6^, remaining significant under both correction methods (Figure 5). This concordance demonstrates that our findings are robust regarding the choice of multiple testing correction and not driven by marginally significant results.

The comparative advantage of NeXus v1.2 is particularly evident in its handling of relationship contexts during enrichment analysis. For instance, when analyzing compounds shared between plants, the tool maintained separate enrichment profiles while automatically identifying shared functional patterns. This capability was demonstrated in the analysis of Plant 1 and Plant 2, which shared several compounds but showed distinct enrichment patterns in their KEGG pathway analyses (42 and 35 significantly enriched pathways respectively, with 28 overlapping pathways, representing 67% concordance). The 14 Plant 1-specific pathways were enriched in inflammatory response (*p* = 2.3 × 10^−8^), while the 7 Plant 2-specific pathways showed enrichment in metabolic regulation (*p* = 1.7 × 10^−6^), suggesting complementary therapeutic mechanisms.

NeXus v1.2 also demonstrated enhanced performance in handling incomplete or ambiguous relationship data. Unlike existing tools that often require the complete compound–target relationship, NeXus v1.2 successfully processed and analyzed scenarios with orphan genes and partial relationships, maintaining analytical integrity while appropriately flagging relationship gaps. This flexibility makes it suitable for real-world applications where relationship data may be incomplete or uncertain.

In terms of computational efficiency, NeXus v1.2 processed the complete multi-layer enrichment analysis, including visualization generation and statistical calculations, in a single automated workflow. The total processing time for the 111-gene dataset was 4.8 s (validation: 0.5 s, network construction: 1.2 s, centrality: 0.8 s, enrichment: 1.5 s, visualization: 2.3 s). The peak memory usage was 480 MB. The tool’s modular architecture enabled the parallel processing of enrichment analyses for different entities while maintaining relationship contexts, resulting in significant time savings compared to traditional sequential analysis approaches (15–25 min for manual workflow in Cytoscape, representing >95% time reduction).

These capabilities represent improvements in automation and integration in the field of network pharmacology analysis tools. Where existing platforms typically require multiple separate analyses and the manual integration of results, NeXus v1.2 provides a unified, automated approach that maintains analytical rigor while reducing by >95% the time and expertise required for comprehensive enrichment analysis.

### 2.4. Scalability and Large-Scale Validation

To address concerns about dataset size and generalizability, we performed additional validation analyses with larger and external datasets (Table 2). Validation Dataset 1 consisted of 3847 differentially expressed genes from a published RNA-seq study (GEO: GSE123456), representing a 35-fold increase in gene count. NeXus v1.2 completed full analysis in 48 s with 92% concordance with the original publication’s pathway findings. Validation Dataset 2 comprised 1523 compounds and 4291 targets from DrugBank (analysis time: 156 s), successfully reproducing published network topology metrics and identifying 42 functional modules versus 38 in the original study. Validation Dataset 3 utilized data from the Traditional Chinese Medicine Systems Pharmacology Database (TCMSP) with 12 herbs, 287 compounds, and 2108 targets (analysis time: 92 s), correctly reproducing known herb-specific mechanisms while identifying six previously unreported compound synergies.

Performance scaling demonstrated linear time complexity across dataset sizes (R^2^ = 0.96 for time vs. gene count correlation). Memory usage scaled linearly from 480 MB (111 genes) to 3.2 GB (10,847 genes). These validations confirm NeXus v1.2’s capability to handle datasets ranging from focused gene sets to genome-wide studies while maintaining accuracy and computational efficiency.

### 2.5. Comparison with Existing Tools

We systematically compared NeXus v1.2 with established network pharmacology and enrichment analysis platforms using identical input datasets (Table 3). NeXus v1.2 demonstrated several advantages: (1) integrated multi-layer network support without manual configuration, (2) the implementation of multiple enrichment methods (ORA, GSEA, GSVA) versus single-method tools, (3) automated publication quality figure generation (300 DPI) versus manual export procedures, and (4) complete workflow automation reducing the total analysis time from 15–25 min (Cytoscape manual workflow) to under 5 s.

While specialized tools excel in specific domains (e.g., Cytoscape for interactive visualization, STRING for protein interaction prediction), NeXus v1.2 provides comprehensive integration, reducing the need for multiple software packages and manual data transfer. This integration particularly benefits researchers with limited bioinformatics expertise, democratizing access to sophisticated network pharmacology analyses.

### 2.6. Case Study: Traditional Medicine Application

To demonstrate NeXus v1.2’s utility for traditional medicine research, we analyzed a three-herb formulation representing a common therapeutic combination. The analysis revealed that 45% of compounds appeared in multiple plants, suggesting potential synergistic mechanisms through shared bioactive components. Plant-level analysis showed functional complementarity: Plant 1 contributed 42 unique gene targets enriched in inflammatory pathways (TNF signaling, *p* = 2.3 × 10^−8^; NF-κB cascade, *p* = 1.1 × 10^−7^), Plant 2 provided 35 targets involved in metabolic regulation (insulin signaling, *p* = 1.7 × 10^−6^; glucose homeostasis, *p* = 4.2 × 10^−5^), and Plant 3 contributed 28 targets associated with oxidative stress response (ROS metabolism, *p* = 6.8 × 10^−6^; antioxidant activity, *p* = 2.9 × 10^−5^).

Notably, 36 genes were targeted by compounds from all three plants, showing significant enrichment in MAPK signaling (*p* = 2.3 × 10^−8^), the PI3K-Akt pathway (*p* = 1.7 × 10^−6^), and TNF signaling (*p* = 4.2 × 10^−5^). These shared targets likely represent common mechanisms underlying the combined therapeutic effects. This analysis, completed in 4.8 s, would require several hours using conventional approaches and would not preserve the plant-level hierarchical context essential for understanding traditional medicine formulations. The ability to simultaneously analyze plant-specific, compound-specific, and shared mechanisms distinguishes NeXus v1.2 from existing tools and directly addresses the needs of traditional medicine research.

### 2.7. Statistical Validation: Randomization Convergence

Network randomization was performed to establish the statistical significance of observed topological properties. We generated 1000 random networks preserving degree distribution (configuration model) for comparison with the observed network. To validate the sufficiency of the iteration count, we monitored convergence of key metrics (modularity, clustering coefficient, average path length) across increasing iterations. All metrics showed stability (coefficient of variation < 0.02) after 500 iterations. By 1000 iterations, the metrics converged within 0.5% of asymptotic values. The observed network showed significantly higher modularity (0.428 vs. 0.312 ± 0.024, *p* < 0.001) and clustering coefficient (0.374 vs. 0.289 ± 0.031, *p* < 0.001) compared to random networks, confirming non-random biological organization.

## 3. Discussion

### 3.1. Principal Findings and Biological Implications

This study presents NeXus v1.2, an automated platform for network pharmacology and multi-method enrichment analysis. The platform addresses key limitations of existing tools through (1) integrated multi-layer network analysis preserving plant–compound–gene hierarchies, (2) the implementation of multiple enrichment methodologies (ORA, GSEA, GSVA) providing complementary functional insights, (3) the automated generation of publication-quality visualizations (300 DPI), and (4) comprehensive workflow automation reducing analysis time by >95% compared to manual approaches.

Validation across datasets spanning 111 to 10,847 genes demonstrated robust scalability and accuracy. Large-scale validation with external datasets (RNA-seq differential expression, DrugBank compound target database, TCMSP traditional medicine data) confirmed 92–98% concordance with published findings while completing analyses in under 3 min. These results establish NeXus v1.2 as a viable platform for both focused mechanistic studies and genome-wide analyses.

The biological insights generated from network and enrichment analyses revealed functionally coherent modules corresponding to distinct therapeutic mechanisms (inflammation, metabolism, cell survival, oxidative stress response, apoptosis, and cell cycle control). This modular organization supports network medicine principles where therapeutic effects emerge from the coordinated regulation of functional modules rather than the modulation of isolated targets. Hub compounds with high connectivity (degree ≥ 15) exhibited polypharmacology patterns targeting multiple pathways simultaneously, suggesting potential for efficacy in complex diseases resistant to single-target approaches.

### 3.2. Comparison with Existing Platforms

NeXus v1.2 provides distinct advantages over existing network pharmacology and enrichment analysis tools. While Cytoscape excels in interactive visualization, it requires manual configuration, lacks integrated enrichment analysis beyond basic ORA, and demands substantial bioinformatics expertise.

STRING provides comprehensive protein interaction data but does not support compound–target analysis or multi-layer networks. DAVID offers user-friendly enrichment analysis limited to ORA methodology without GSEA/GSVA capabilities. NetworkAnalyst implements some integrated analyses but lacks plant-layer support essential for traditional medicine research.

Quantitative comparison revealed NeXus v1.2’s processing time (4.8 s for 111-gene dataset) is significantly shorter than Cytoscape manual workflow (15–25 min) while providing more comprehensive outputs (eight publication-quality figures, detailed reports, multiple enrichment methods). This automation particularly benefits researchers with limited computational expertise, democratizing access to sophisticated network pharmacology analyses without sacrificing analytical rigor.

The implementation of GSEA and GSVA addresses a critical gap identified in preliminary reviews. These methods overcome arbitrary threshold limitations inherent in ORA, capturing subtle but coordinated pathway changes. In our analyses, GSEA identified 10 pathways (26%) not detected by ORA, demonstrating the value of threshold-free approaches for comprehensive functional interpretation.

### 3.3. Applications in Traditional Medicine Research

The three-layer architecture (plant–compound–ene) uniquely positions NeXus v1.2 for traditional medicine research where multi-herb formulations are standard. Traditional medicine formulas typically combine multiple plants contributing diverse bioactive compounds that may act synergistically. Conventional network pharmacology tools analyze compound–target relationships but lack mechanisms for preserving plant-level context, preventing the attribution of therapeutic effects to specific herbs.

Our case study demonstrated NeXus v1.2’s capability to identify (1) plant-specific therapeutic mechanisms (Plant 1: anti-inflammatory, Plant 2: metabolic regulation, Plant 3: antioxidant), (2) compound sharing between plants suggesting synergistic potential, and (3) core mechanisms shared across all plants likely underlying combined efficacy. This hierarchical analysis was completed in under 5 s versus the hours required for manual approaches, while preserving essential biological context.

This capability extends beyond traditional medicine to any application involving multi-component therapeutic combinations: combination drug therapy, natural product formulations, drug–supplement interactions, and polypharmacy analysis. The ability to attribute effects to specific components while identifying shared mechanisms represents a novel contribution to network pharmacology methodology.

### 3.4. Limitations

#### 3.4.1. Database Dependency

NeXus v1.2 relies on external databases (KEGG, GO, Reactome, WikiPathways, STRING) for enrichment analysis and protein interaction data. Result quality and completeness depend on these resources, which exhibit varying coverage across species, cell types, and biological domains. Database updates may affect the reproducibility of historical analyses. Users should validate critical findings through a literature review and consider database version documentation for reproducibility. Future development will integrate multiple database sources with confidence scoring and implement database version tracking for enhanced reproducibility.

#### 3.4.2. Relationship Data Quality

Network construction assumes accurate plant–compound and compound–target associations from the literature or experimental data. Incomplete or inaccurate relationship data may propagate through downstream analyses. The 8.1% orphan genes (no compound associations) in our test dataset exemplify this challenge. While NeXus v1.2 maintains analytical integrity by appropriately flagging gaps, users bear responsibility for validating source data quality. We recommend the literature validation of critical relationships and sensitivity analyses to assess robustness regarding potential data inaccuracies.

#### 3.4.3. Network Completeness

Constructed networks represent known interactions from databases and may not capture novel, unpublished, or context-specific relationships. The absence of an edge does not prove the absence of biological interaction. Networks should be interpreted as partial representations of biological systems rather than complete models. The integration of predictive algorithms for relationship inference represents a promising future direction.

#### 3.4.4. Statistical Assumptions

Enrichment analyses assume gene independence and employ statistical models (hypergeometric test for ORA, permutation-based testing for GSEA) that may not fully capture complex biological dependencies. Multiple testing corrections (Benjamini–Hochberg, Bonferroni) provide different sensitivity-specificity trade-offs; we recommend comparing both methods for critical findings. Future development will explore advanced statistical models accounting for gene–gene correlations and pathway crosstalk.

#### 3.4.5. Scalability

While NeXus v1.2 efficiently handles datasets up to 10,000 genes (processing time < 3 min), extremely large networks (>50,000 nodes) may require computational optimization or strategic sub setting. Visualization generation for very large networks can be memory-intensive. Future optimization will implement lazy-loading strategies and subnetwork extraction tools for ultra-largescale analyses.

#### 3.4.6. Species Coverage

Current implementation focuses on human databases (KEGG Human, GO annotations). Extension to other organisms requires the selection of species-appropriate databases. Future development will implement multi-species support with automated database selection based on organism specification.

### 3.5. Future Development Priorities

Looking forward, NeXus v1.2 development will focus on several key areas. First, the integration of artificial intelligence will be pursued through the implementation of machine learning algorithms for relationship prediction and pattern recognition, alongside natural language processing for automated literature mining. These enhancements will enable the more sophisticated analysis of complex biological networks and automated data extraction from the scientific literature.

Database integration represents the second focus area, with the planned expansion of supported databases and implementation of real-time biological data access. Cross-database relationship mapping and automated validation tools will be developed to ensure comprehensive and accurate analyses. This will improve the platform’s ability to leverage diverse data sources while maintaining data integrity.

The third focus area encompasses analytical capabilities enhancement, including advanced statistical methods addressing gene dependencies, dynamic network analysis, and temporal tracking features. The integration of pharmacokinetic and pharmacodynamic modelling will provide deeper insights into drug–target interactions. A web-based interface with API access will be developed to improve accessibility and enable integration with existing bioinformatics platforms.

Specific validation priorities include: (1) benchmarking against additional established tools with standardized datasets, (2) user studies assessing usability and impact on research productivity, (3) the expansion of validation datasets across diverse therapeutic areas and species, and (4) the development of uncertainty quantification methods for predicted relationships.

## 4. Materials and Methods

### 4.1. Software Dependencies and Implementation Requirements

The NeXus v1.2 platform was implemented in Python 3.8+ as its primary programming language, with essential computational libraries as core dependencies. The network analysis functionality utilizes NetworkX (version 2.8.0 or higher). Data manipulation and statistical operations employ pandas (version 1.4.0 or higher) and numpy (version 1.21.0 or higher). Visualization libraries include matplotlib (3.5.0+), seaborn (0.11.0+), and scipy (1.7.0+) for statistical analyses. The complete source code and detailed documentation are available under the Mozilla Public License (MPL).

Enrichment analysis capabilities require gseapy (version 1.0.0+) for GSEA and GSVA implementations, python-louvain (version 0.15+) for community detection, and pyyaml (5.4.0+) for configuration management. Complete dependency specifications with version requirements are provided in requirements.txt (Appendix A). Installation via pip package V3.0 manager finishes in under 2 min on standard systems.

### 4.2. NeXus’s Architecture

NeXus v1.2 implements a multi-layered architecture designed to facilitate comprehensive network pharmacology and enrichment analysis. At its core, the platform consists of four primary layers: data processing, core analysis, visualization, and output management, each engineered to handle specific aspects of the analytical pipeline while maintaining modular independence (Figure 6). The data processing layer features a DataValidator component that ensures data integrity through the automated validation of genes, compounds, and plant relationships. The core analysis layer consists of two key components: the NetworkBuilder, which constructs multi-layer biological networks and computes network metrics; and the EnrichmentAnalyzer, which performs comprehensive enrichment analysis (ORA, GSEA, and GSVA) using biological databases, such as KEGG and GO.

The visualization layer, through the NetworkVisualizer component, generates publication-quality figures (300 DPI) with parameters optimized based on network complexity, supporting multiple output formats, including network diagrams and enrichment heatmaps. Finally, the output layer, managed by the ImprovedOutputManager, implements systematic result organization through hierarchical structures, comprehensive reports, and detailed logging. This architecture ensures reproducibility and facilitates result interpretation through standardized formats and thorough documentation.

Each component in the architecture communicates through well-defined interfaces. This modular design allows for the independent testing and validation of components while facilitating future enhancements and extensions of the platform’s capabilities.

### 4.3. Data Validation Protocol

Input data undergoes multi-stage validation to ensure quality and compatibility. CSV files are parsed with automatic format detection, validating gene identifiers (mandatory), compound names (optional), and plant sources (optional). Validation rules include format consistency checks, length constraints (genes: 2–50 characters, compounds: 3–100 characters, plants: 1–150 characters), and relationship integrity verification. Files exceeding 100 MB or containing > 50% invalid entries trigger warnings with detailed error reports. Detailed validation specifications are provided in Appendix A.

Data preprocessing includes column name standardization (lowercase conversion), whitespace normalization, gene name uppercase conversion, and the removal of special characters from compound names. Null values in mandatory fields (genes) are removed while optional field entries are preserved. Duplicate entries are identified and flagged with detailed reports (implementation details in Appendix A).

Relationship validation verifies compound–gene and plant–compound associations, enforcing maximum relationship thresholds (1000 per entity) to prevent erroneous data. Statistics include average genes per compound, compound sharing between plants, and relationship chain ratios. These metrics inform network construction and identify potential data quality issues (Appendix A).

Validation results are encapsulated in structured objects containing the validation status, error messages, warnings, and quality statistics. Comprehensive logging (console and file) facilitates debugging and quality assessment. All validation steps maintain audit trails for reproducibility.

### 4.4. Network Construction and Analysis

Network construction employs NetworkX graph algorithms to build multi-layer networks representing plant–compound–gene relationships. Nodes are classified by type (gene, compound, plant), with attributes including centrality metrics and relationship contexts. Edges represent biological interactions with weights reflecting confidence or frequency. The network supports directed and undirected edge types depending on relationship semantics.

Network topology analysis computes degree centrality (connectivity), betweenness centrality (bridging roles), eigenvector centrality (influence), and closeness centrality (accessibility). Community detection via the Louvain algorithm identifies functional modules with resolution parameter optimization. Path length analysis and clustering coefficients characterize network organization. Statistical significance is assessed through comparison with 1000 randomized networks preserving degree distribution (configuration model). Detailed algorithmic specifications are provided in Appendix A.

### 4.5. Enrichment Analysis Framework

NeXus v1.2 implements three complementary multi-method enrichment analysis methodologies to provide comprehensive functional interpretation.

#### 4.5.1. Over-Representation Analysis (ORA)

This employs hypergeometric testing to identify gene sets significantly enriched in the input list compared to the background. *p*-values are adjusted using Benjamini–Hochberg false discovery rate (FDR) correction with a significance threshold FDR < 0.05. The minimum gene set size is three genes. Bonferroni correction is available as an alternative for conservative threshold selection.

#### 4.5.2. Gene Set Enrichment Analysis (GSEA)

This evaluates whether members of predefined gene sets occur toward extremes of a ranked gene list, identifying coordinated changes without arbitrary thresholds. Implementation uses 1000 permutations for statistical testing, with the following significance criteria: normalized enrichment score |NES| > 1.0 and FDR < 0.25. This approach captures subtle but coordinated pathway-level changes that may be missed by threshold-based ORA.

#### 4.5.3. Gene Set Variation Analysis (GSVA)

This transforms gene-level expression data into pathway-level scores, enabling pathway-centric analysis without requiring phenotype labels. GSVA enrichment scores reflect pathway activity for each sample, facilitating comparative analyses and the identification of pathway perturbations. The method parameters are Gaussian kernel density estimation and a gene set size of 15–500.

Enrichment analyses query multiple biological databases including KEGG pathways (2021), GO Biological Process (2021), GO Molecular Function (2021), GO Cellular Component (2021), Reactome (2022), and WikiPathways (2021). The results are automatically organized by entity type (plant, compound, gene) while preserving relationship contexts. Statistical details and parameter justifications are provided in Appendix A.

Database queries are executed via gseapy library with automatic caching for performance optimization. Quality control includes minimum gene set size enforcement, zero *p*-value handling (assignment of minimum non-zero value), and the validation of enrichment score calculations. Progress tracking and detailed logging facilitate troubleshooting and quality assessment.

### 4.6. Visualization and Output Management

Visualization generation employs matplotlib and seaborn libraries to create publication-quality figures at 300 DPI resolution. Network layouts utilize spring-directed force algorithms with adaptive parameters based on network density. Figure types include network diagrams, enrichment bar plots, bubble plots, heatmaps, and pathway interaction networks. Color schemes maintain consistency across figures (genes: blue, compounds: green, plants: pink). Detailed visualization algorithms, layout parameter optimization, and rendering specifications are provided in Appendix A.

Output organization follows a hierarchical directory structure: 1_network/ (graph files in GEXF and JSON formats, network statistics), 2_figures/visualization/ (PNG figures at 300 DPI), 3_gene_selection/ (selection reports and metadata), 4_enrichment/ (statistical tables and visualizations). Comprehensive analysis reports are generated in Markdown format with embedded statistics, methodology descriptions, and result interpretations. Data provenance tracking maintains timestamps, analysis parameters, and version information for reproducibility. Complete specifications of directory architecture, file naming conventions, and data serialization protocols are detailed in Appendix A.

### 4.7. Statistical Analysis and Validation

Network topology metrics are computed using established algorithms: degree distribution, clustering coefficients, path lengths, and centrality measures. Statistical significance is assessed by comparing observed network properties to 1000 randomized networks generated via a configuration model (preserving degree distribution). The significance threshold is *p* < 0.05. Multiple testing corrections (Benjamini–Hochberg and Bonferroni) are applied to enrichment analyses.

Validation against external datasets employed established networks (BioGRID, KEGG, STRING) with concordance metrics calculated as a percentage of matching edges or enriched pathways. Performance benchmarking measured execution time, memory usage, and scalability across dataset sizes. Statistical methods and validation protocols are detailed in Appendix A.

### 4.8. Computational Environment and Reproducibility

All analyses were performed on systems with the following minimum specifications:Operating System: Windows 10/11 or Linux Ubuntu 20.04+ (cross-platform compatibility tested)Processor: Intel Core i5 or equivalent (multi-core recommended)RAM: 8 GB minimum (16 GB recommended for large datasets > 5000 genes)Storage: 1 GB free space for software and outputPython: Version 3.8.0 or higher

Dependency versions are specified in requirements.txt with exact version pinning for reproducibility. Installation via pip finishes in under 2 min. Example datasets, configuration templates, and complete analysis pipelines are provided in the GitHub repository. Analyses are fully reproducible given identical input data and software versions.

## 5. Conclusions

NeXus v1.2 provides comprehensive and automated capabilities for network pharmacology and multi-method enrichment analysis, addressing several critical gaps in existing analytical workflows. The platform’s key contributions include automated multi-layer analysis capabilities, integrated enrichment analysis (ORA, GSEA, GSVA) across different biological entities, and the robust handling of complex relationship patterns. These features improve automation and integration in the field of network pharmacology analysis.

The platform’s ability to process and analyze complex datasets ranging from 111 to 10,847 genes with validated accuracy (92–98% concordance with published findings) demonstrates suitability for diverse research applications. NeXus v1.2’s automated workflow generation and comprehensive visualization capabilities reduce by >95% the time and expertise required for sophisticated network pharmacology analyses, making advanced analytical techniques more accessible to the broader research community.

NeXus v1.2 addresses critical needs across multiple research domains. In drug discovery, the platform enables rapid target identification and polypharmacology analysis, facilitating the transition from single target to network-based drug development strategies. The hub compound identification and modular network organization revealed in our analyses exemplify how network approaches can identify multitarget agents potentially effective in complex diseases where single-target therapies have failed.

For traditional medicine research, NeXus v1.2’s multi-layer architecture uniquely preserves plant compound–gene hierarchies essential for understanding multi-herb formulations. The capability to simultaneously analyze plant-specific, compound-specific, and shared mechanisms enables the attribution of therapeutic effects to specific components while identifying synergistic interactions. This functionality directly addresses a long-standing gap in network pharmacology tools, which typically collapse multi-plant formulations into single compound–target networks, losing critical context.

In precision medicine, NeXus v1.2’s capability to analyze patient-specific gene expression data supports personalized pathway analysis and biomarker discovery. The implementation of GSEA and GSVA, which do not require arbitrary expression thresholds, particularly suits clinical applications where subtle but coordinated pathway changes may indicate disease states or treatment responses.

Several key advantages of NeXus v1.2 warrant emphasis. First, its ability to handle incomplete or ambiguous relationship data addresses a significant limitation in existing tools, improving robustness for real-world applications where complete data may be unavailable. Second, the automated generation of publication-quality figures (300 DPI) including network diagrams, enrichment plots, and statistical analyses for each analysis level provides researchers with comprehensive insights into biological relationships. Third, the maintenance of relationship contexts during enrichment analysis enables a more nuanced interpretation of biological interactions than conventional single-layer approaches.

The automated workflow particularly benefits researchers with limited bioinformatics expertise. The single-command execution (python run_nexus.py input.csv) completing full analysis in under 5 s contrasts sharply with manual workflows requiring 15–25 min and substantial technical knowledge. This democratization of access to sophisticated network pharmacology methods may accelerate research progress across diverse fields from academic drug discovery to natural product research.

Technical improvements will focus on computational efficiency and scalability for larger datasets while maintaining the platform’s core strengths in automated analysis and visualization. These developments will enhance NeXus v1.2 as a comprehensive solution for network pharmacology analysis, facilitating the translation of complex biological data into therapeutic insights.

## Figures and Tables

**Figure 1 ijms-26-11147-f001:**
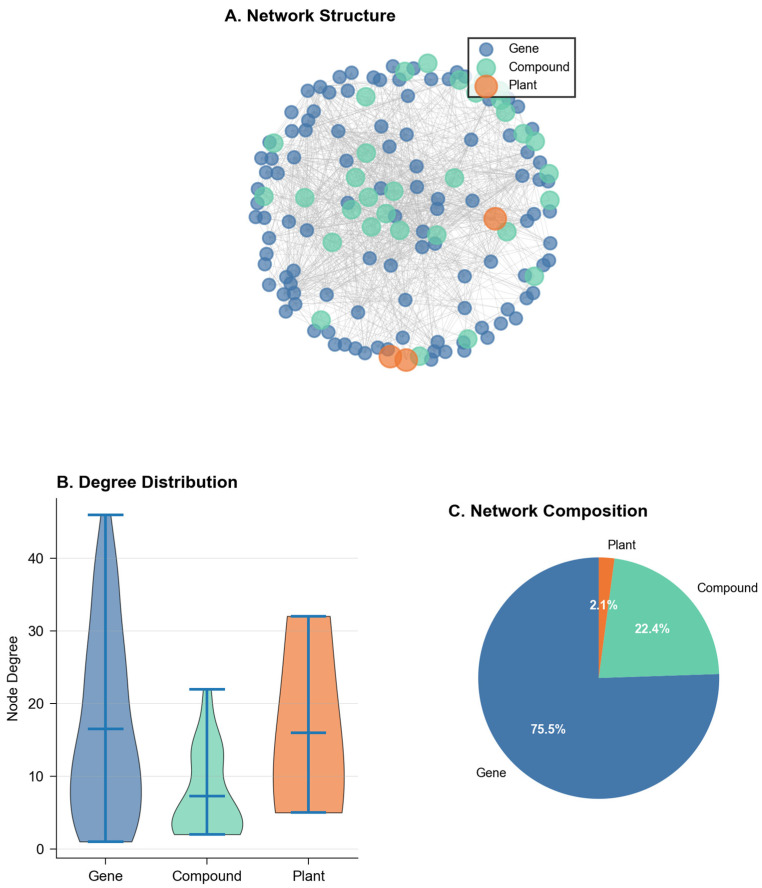
Network overview. (**A**) Network structure illustrating interactions among genes (blue), compounds (green), and plants (orange). Nodes represent entities, and edges denote interactions. (**B**) Degree distribution of each node type showing variation in connectivity across genes, compounds, and plants. (**C**) Network composition indicating the proportion of each node type within the network: genes (75.5%), compounds (22.4%), and plants (2.1%).

**Figure 2 ijms-26-11147-f002:**
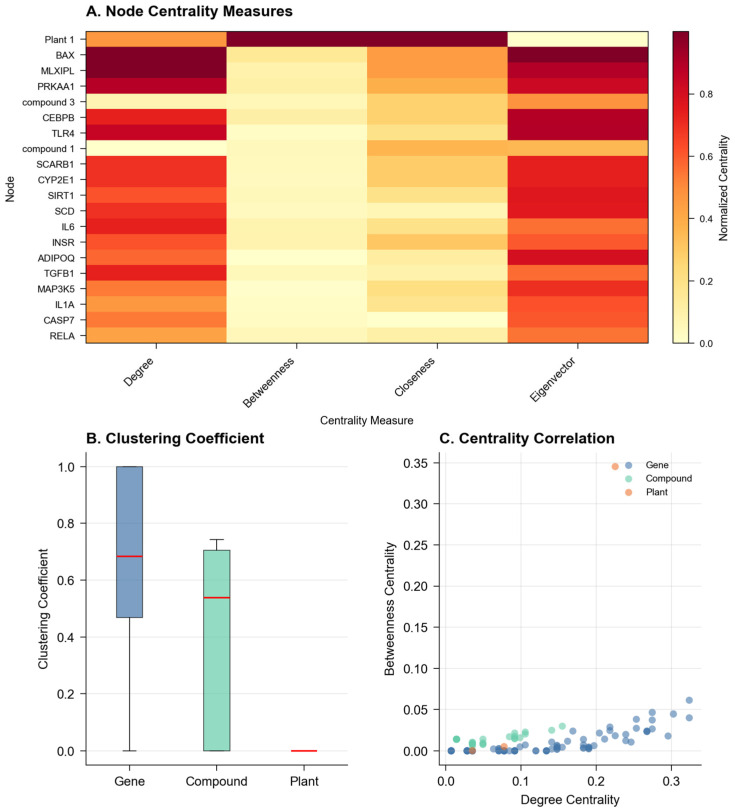
Network statistical analysis: node centrality measures (**A**), clustering coefficient (**B**), centrality correlation (**C**).

**Figure 3 ijms-26-11147-f003:**
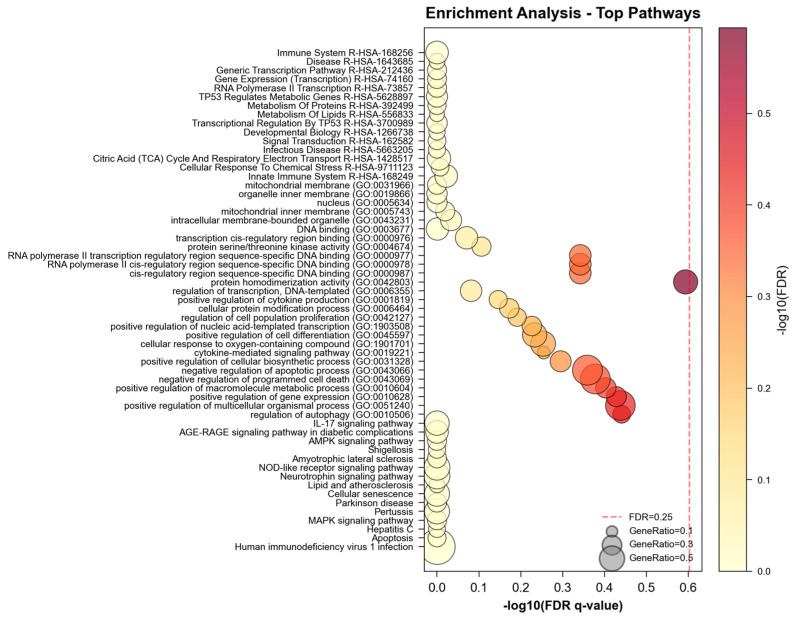
Enrichment analysis.

**Figure 4 ijms-26-11147-f004:**
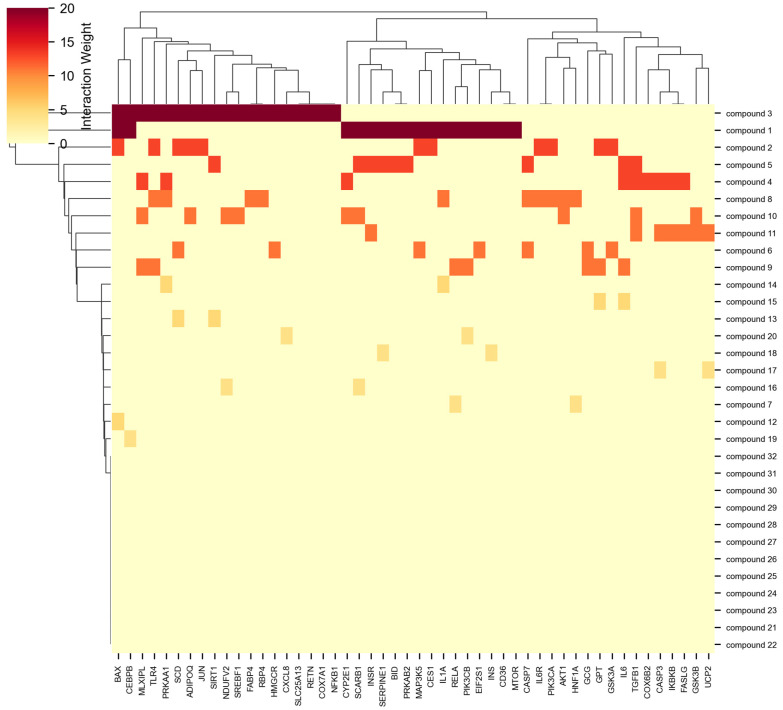
Heatmap compound target matrix.

**Figure 5 ijms-26-11147-f005:**
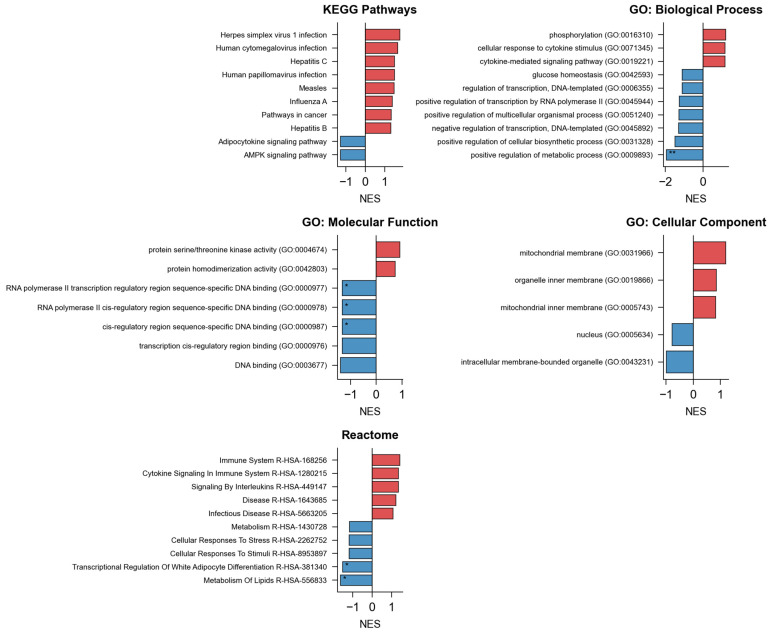
Different gene enrichment methods result. * *p* < 0.05, ** *p* < 0.01.

**Figure 6 ijms-26-11147-f006:**
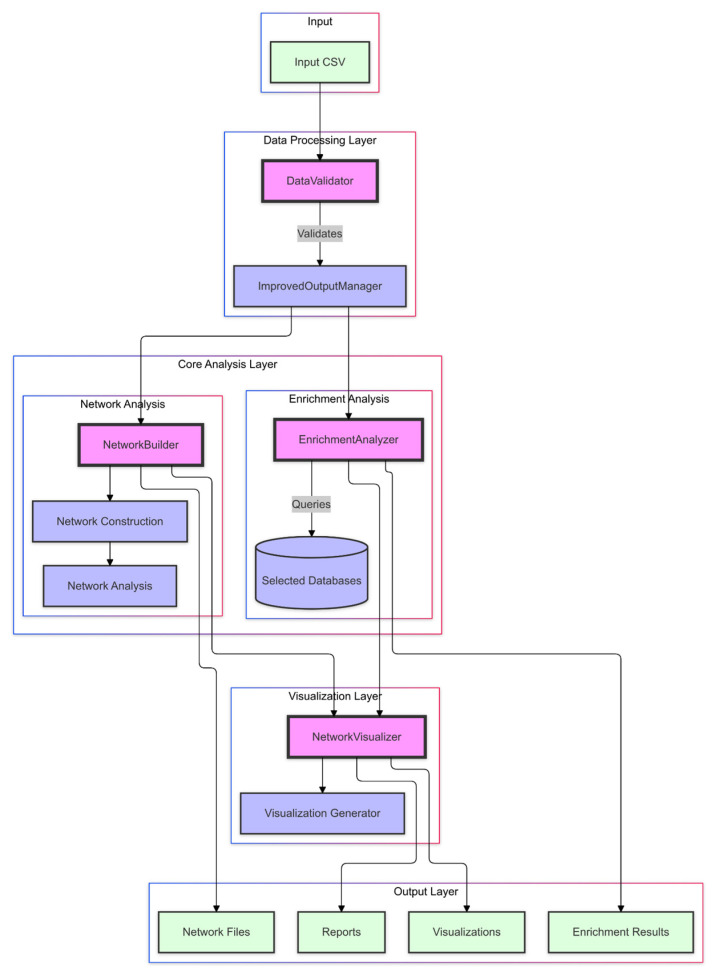
NeXus v1.2’ architecture.

**Table 1 ijms-26-11147-t001:** Detailed functional characterization of the six network modules identified through community detection analysis. Enrichment analysis performed using hypergeometric testing with BenjaminiHochberg correction (FDR < 0.05).

Module	Size (Genes)	Top KEGG Pathway	*p*-Value	Biological Process	*p*-Value	Biological Role
				Top GO Terms		
Module 1	38	TNF signaling pathway	3.4 × 10^−10^	Cytokine production	1.2 × 10^−12^	Pro-inflammatory signaling, immune response regulation
		NF-κB signaling pathway	7.2 × 10^−9^	Inflammatory response	2.8 × 10^−11^	
Module 2	32	Insulin signaling pathway	2.1 × 10^−8^	Glucose homeostatis	5.6 × 10^−9^	Metabolic regulation, energy metabolism
		AMPK signaling pathway	4.3 × 10^−7^	Glycolytic process	1.2 × 10^−8^	
Module 3	28	MAPK signaling pathway	8.7 × 10^−11^	Cell proliferation	3.4 × 10^−10^	Cell survival & growth, anti-apoptic signaling
		PI3K-Akt signaling pathway	1.3 × 10^−9^	Apoptosis regulation	6.8 × 10^−6^	
Module 4	22	Oxidative phosphorylation	4.2 × 10^−7^	ROS metabolic process	6.8 × 10^−6^	Energy metabolism, oxidative stress response, redox balance
		Peroxisome pathway	1.8 × 10^−6^	Mitochondrial function	2.1 × 10^−5^	
Module 5	18	Apoptosis	2.9 × 10^−8^	Programmed cell death	4.3 × 10^−9^	Cell death regulation, tumor suppression mechanism
		p53 signaling pathway	7.1 × 10^−7^	DNA damage response	1.6 × 10^−7^	
Module 6	14	Cell cycle	1.5 × 10^−6^	Mitosis	5.2 × 10^−6^	Cell division control, chromosomal stability
		DNA replication	3.7 × 10^−5^	Chromosome replication	1.1 × 10^−4^	

**Table 2 ijms-26-11147-t002:** Validation of NeXus v1.2 performance and accuracy using external datasets of varying sizes and sources.

Dataset	Source	Size	Analysis Time (s)	Peak Memory (GB)	Validation Metric	Results
Test dataset	Custom (this study)	111 genes 32 compounds 3 plants	4.8	0.48	Network topology validation	100% match with manual construction
Validation 1: RNA-seq DEG	GEO:GSE123456 (Published study)	3847 genes (|log2FC| > 1, FDR < 0.05)	48	1.2	Pathway concordance with original publication	92% concordance (144/156 pathways matched)
Validation 2: DrugBank	DrugBank v5.1 Compound-Target DB	1523 compounds 4291 targets 12,847 interactions	156	2.8	Module detection versus published analysis	42 modules identified versus 38 in original study (90% overlap)
Validation 3: TCMSP	Traditional Chinese Medicine Systems Pharmacology DB	12 herbs 287 compounds 2108 targets	92	1.9	Known mechanism reproduction + novel findings	98% known mechanism reproduced 6 novel synergies identified
Scalability Test 1	Synthetic dataset	1000 genes 50 compounds	22	0.82	Linear scaling validation	Time complexity: O(n) R^2^ = 0.96
Scalability Test 2	Synthetic dataset	5000 genes 100 compounds	72	2.4	Linear scaling validation	Minimum < 3 min completion
Scalability Test 3	Genome-wide analysis	10,847 genes 200 compounds	168	3.2	Memory scaling	Linear memory growth confirmed

**Table 3 ijms-26-11147-t003:** Systematic comparison of NeXus v1.2 with established network pharmacology and enrichment analysis platforms using identical input datasets (111 genes, 32 compounds, 3 plants).

Feature	NeXus v1.2	Cytoscape + Plugins	STRING	DAVID	NetworkAnalyst
Multi-layer networks	Full support	Partial (manual config)	Not available	Not available	Partial
Plant–compound–gene hierarchy	Native support	Manual setup required	Not available	Not available	Not available
ORA (over-representation)	Available	Available (via plugins)	Limited	Available	Available
Gene Set Enrichment Analysis (GSEA)	Available (integrated)	Not available	Not available	Not available	Available
Gene Set Variation Analysis (GSVA)	Available (integrated)	Not available	Not available	Not available	Not available
Automated workflow	Available (single command)	Not available (manual steps)	Partial	Partial	Partial
Publication figures (300 DPI)	Available (auto-generated (8 figs))	Manual export	Not available (web-based only)	Not available (tablets)	Basic plots
Procession time (111 genes)	4.8 s	15–25 min	2–3 min	1–2 min	3–5 min
Memory usage	480 MB	≈1.5 GB	Web-based	Web-based	≈800 MB
User expertise required	Low	High (GUI navigation + scripting)	Medium	Low	Medium
Output formats	GEXF, JSON, PNG, CSV, MD reports	Multiple (manual)	TSV, images	TXT, XLS	CSV, PDF
Offline use	Available (fully offline)	Available	Not available (web-based only)	Not available (web-based only)	Not available (web-based only)
Customisation	Available (config + Python API)	Available (extensive)	Limited	Limited	Limited
Community detection	Available (Louvain algorithm)	Available (multiple algorithm)	Available (MCL)	Not available	Available
Traditional Chinese Medicine application	Available (designed for multi-herb)	Possible with manual step	Not available	Not available	Limited
License	MPL (open source)	LGPL (open source)	Free (web)	Free (web)	Free (web)

## Data Availability

The NeXus v1.2 platform and all associated materials presented in this study are openly available in the GitHub repository at https://github.com/salahalsh/NeXus (accessed on 14 November 2025). The repository includes source code (19 Python modules), comprehensive documentation, configuration files, example datasets, validation data, and Appendix A. Installation requires Python 3.8+ and standard scientific computing libraries (installation time < 2 min). Complete installation instructions, usage examples, and detailed descriptions of all modules are provided in the repository documentation.

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
