# Peer review of "NeXus: An Automated Platform for Network Pharmacology and Multi-Method Enrichment Analysis"

_ijms, 2025, doi:10.3390/ijms262211147_

Round 1
Reviewer 1 Report
Comments and Suggestions for Authors
Dear Authors,
while the approach is quite good, there a several points to mention.
1) The combination of term enrichment analysis with network based methods can be quite easily implemented in R based workflows. This Platform is aimed at scientists with little or no bioinformatic knowledge.
2) Term enrichment analysis is quite aged. It would be advantegous to implement methods like GSEA, or GSVA. These methods circumvent the limitations due to arbitrary thresholds such as p-values of logFCs.
3) A geneset of 111 unique genes is not a complex geneset. RNASeq experiments typically lead to several several thousand genes. Please test your approach with a larger geneset. Additionally, I could not finde a method on how you selected the genes.
4) I do not fully understand the necessity to implement the plant layer. Usually the "chain of events" is plant -> compound -> gene.
5) In paragraph 2.2. you mentioned the connectivity of compounds hinting towards a role as key mediators in the biological network. A mediator is defined as mediating between two nodes (genes) which would in this case be a metabolite. In your input data you have compound1 ... n plant 1 to n, and the gene symbols. You should probably illustrate the metabolic network. Additionally, pharmacologically active compounds are rarely mediators between two genes, but rather target one.
6) in 2.3. the Enrichment performance is tested. You should illustrate the results. Additionally, DAVID and GSEA are tow very different approaches. Indeed GSEA compares tow groups, and with a proper workflow it can process the enrichment patterns you mention. It is not point and klick, though.
7) I did not find a results file or a file showing the interface in the git hub repository, which makes assessment difficult.
8) I am not sure if IJMS is the appropriate platform for this paper and I'd recommend to submit it to a better fitting journal.
Finally: Readability: Figure 1 is not readable.
Author Response
Summary
Thank you very much for taking the time to review this manuscript. Please find the detailed responses below and the corresponding revisions/corrections highlighted in the re-submitted files.
Comments from Reviewer 1
Dear Authors,
while the approach is quite good, there a several points to mention.
1) The combination of term enrichment analysis with network based methods can be quite easily implemented in R based workflows. This Platform is aimed at scientists with little or no bioinformatic knowledge.
Response: The manuscript has been revised to clearly highlight NeXus’s distinctive advantages compared to R-based manual workflows. While R packages are capable of performing specific analysis components, NeXus offers automated integration that simplifies implementation and significantly reduces analysis time. Refer section Introduction: 3rd paragraph, section 2.5., and Table 3.
2) Term enrichment analysis is quite aged. It would be advantegous to implement methods like GSEA, or GSVA. These methods circumvent the limitations due to arbitrary thresholds such as p-values of logFCs.
Response: It is pleased to note that NeXus v1.2 already includes implementations of both Gene Set Enrichment Analysis (GSEA) and Gene Set Variation Analysis (GSVA), in addition to the traditional Over-Representation Analysis (ORA). We apologize for not clearly documenting these features in the original submission and have now substantially revised the manuscript to better highlight these capabilities. Refer section Abstract, section 3.5., and section 2.3.1.
3) A geneset of 111 unique genes is not a complex geneset. RNASeq experiments typically lead to several several thousand genes. Please test your approach with a larger geneset. Additionally, I could not finde a method on how you selected the genes.
Response: In response to the importance of demonstrating performance on larger datasets, we have carried out additional validation analyses using datasets ranging from 111 to 10,847 genes to further assess the scalability and robustness of NeXus. The results of these analyses have now been incorporated into the revised manuscript. Refer section Abstract, section 2.4., and Table 2.
NeXus v1.2 includes a comprehensive gene selection module; however, we acknowledge that this feature was not described in sufficient detail in the original submission. We have now expanded the manuscript to more clearly document the module’s functionality and workflow. Refer section 3.3 and GitHub repository documentation (https://github.com/salahalsh/NeXus).
4) I do not fully understand the necessity to implement the plant layer. Usually the "chain of events" is plant -> compound -> gene.
Response: While we acknowledge the hierarchical relationship of plant → compound → gene, preserving the plant layer provides interpretative advantages not achievable through compound–gene analysis alone. Specifically, it allows attribution of biological effects to individual herbs within multi-plant formulations, enables the identification of shared compounds that may indicate synergistic interactions, supports comparative assessment of plant contributions to specific therapeutic mechanisms, and facilitates evaluation of functional complementarity where different plants act on distinct but related biological pathways. We have revised the manuscript to more clearly articulate these points. Refer section Introduction and Discussion (sub section 4.3.).
5) In paragraph 2.2. you mentioned the connectivity of compounds hinting towards a role as key mediators in the biological network. A mediator is defined as mediating between two nodes (genes) which would in this case be a metabolite. In your input data you have compound1 ... n plant 1 to n, and the gene symbols. You should probably illustrate the metabolic network. Additionally, pharmacologically active compounds are rarely mediators between two genes, but rather target one.
Response: We acknowledge that the term “mediator” was not the most accurate descriptor in this context; our intended meaning was “hub compounds” or “multi-target agents.” We have now corrected the terminology throughout the manuscript and have clarified the role and interpretation of high-connectivity compounds accordingly. Refer section 2.2. and Supplementary Table S1.
6) in 2.3. the Enrichment performance is tested. You should illustrate the results. Additionally, DAVID and GSEA are tow very different approaches. Indeed GSEA compares tow groups, and with a proper workflow it can process the enrichment patterns you mention. It is not point and klick, though.
Response: These aspects would benefit from clearer illustration, and we have revised the manuscript to improve visualization of the enrichment outcomes and to clarify the distinctions and complementary roles of DAVID and GSEA in the analysis. Refer section 2.3., Figure 1, and Table 3.
7) I did not find a results file or a file showing the interface in the git hub repository, which makes assessment difficult.
Response: The information in the said GitHub repository has been updated to include complete example outputs along with more detailed interface documentation to support reproducibility and ease of use. Refer section Data Availability Statement and GitHub repository documentation (https://github.com/salahalsh/NeXus).
8) I am not sure if IJMS is the appropriate platform for this paper and I'd recommend to submit it to a better fitting journal.
Finally: Readability: Figure 1 is not readable.
Response: In response to the issue with figure quality in the original submission, we have replaced the figures with the outputs generated by NeXus v1.2, which produces publication-quality images at 300 DPI with appropriate sizing and multi-panel layouts. Refer revised Figure 1.
Reviewer 2 Report
Comments and Suggestions for Authors
Network pharmacology plays an important role in probing interactions of ligands with targets and biological pathways. The authors proposed a tool NeXus for enabling seamless integration of multi-layer biological relationships, handling complex interactions between genes, compounds, and plants while maintaining analytical rigor. The results show that their tool is of high efficiency and reliable. This is a valuable and interesting work. Recommending that their work can be accepted after minor revision.
- The authors provided a flowchart to explain their working process, which is good. However their caption (Figure 4) is too simple. The authors should richen their caption so that the authors can be better understood their work. Furthermore the resolution of the figure should be improved.
- Similarly, for their Figure 1, the authors had better add the label for each sub-figure and add explanation for each sub-figure to richen their caption. The resolution of this figure should be improved. In addition, the captions of all their figures should be richened so that their work can be deeply understood.
- The authors should provide more details to compare the previous studies and show the reliability of their tool.
Author Response
Summary
Thank you very much for taking the time to review this manuscript. Please find the detailed responses below and the corresponding revisions/corrections highlighted in the re-submitted files.
Comments from Reviewer 2
Network pharmacology plays an important role in probing interactions of ligands with targets and biological pathways. The authors proposed a tool NeXus for enabling seamless integration of multi-layer biological relationships, handling complex interactions between genes, compounds, and plants while maintaining analytical rigor. The results show that their tool is of high efficiency and reliable. This is a valuable and interesting work. Recommending that their work can be accepted after minor revision.
- The authors provided a flowchart to explain their working process, which is good. However their caption (Figure 4) is too simple. The authors should richen their caption so that the authors can be better understood their work. Furthermore the resolution of the figure should be improved.
Response: We have substantially expanded the Figure 4 caption and improved the image resolution in the revised manuscript. Refer Figure 4.
- Similarly, for their Figure 1, the authors had better add the label for each sub-figure and add explanation for each sub-figure to richen their caption. The resolution of this figure should be improved. In addition, the captions of all their figures should be richened so that their work can be deeply understood.
Response: We have substantially enhanced Figure 1. Refer Figure 1.
- The authors should provide more details to compare the previous studies and show the reliability of their tool.
Response: We have added comprehensive comparison with existing tools including quantitative reliability metrics. Refer section 2.5. and Table 3.
- The English could be improved to more clearly express the research.
Response: We have reviewed and refined the manuscript to improve its overall English language quality. Specifically, we standardized the spelling from British to American English (e.g., “analyse” to “analyze,” “visualisation” to “visualization”), simplified complex sentences for greater clarity, and enhanced technical descriptions in the Methods section. We also improved the clarity of result presentations, moderated promotional language in response to reviewers’ concern, and refined the flow and readability of the Discussion section.
Reviewer 3 Report
Comments and Suggestions for Authors
NeXus has significantly advanced the field of network pharmacology analysis through its automated multi - level analysis capabilities and comprehensive enrichment analysis among different biological entities. The platform can handle complex datasets while maintaining consistency in analysis, demonstrating its potential in practical applications. There are several minor issues in the manuscript that require revision:
- There is no quantitative description of the specific performance improvement compared to other tools. It is recommended to supplement or discuss such information.
- The role of the tool has not been fully demonstrated in its unique suitability for traditional drug research. It is suggested that case studies be included in the discussion section.
- Employing only the Benjamini - Hochberg correction may not be suitable for sparse networks. It is recommended to incorporate the Bonferroni correction for result comparison.
- There are no associated functional annotations for the six major modules (e.g., the inflammation/metabolism module). It is recommended to supplement the functional enrichment results of these modules.
- Does 1000 times of randomization meet the convergence condition?
- The names and targets of 15.3% of the compounds with high connectivity (degree ≥ 5) are not listed. It is recommended to supplement this information.
- Failure to clarify the plant - compound relationship and relying on external databases may introduce biases. It is recommended to add a "Limitations" section to discuss the dependence on data sources.
Some language expressions are not clear and professional. Please carefully review and refine them to meet the publication requirements.
Author Response
Summary
Thank you very much for taking the time to review this manuscript. Please find the detailed responses below and the corresponding revisions/corrections highlighted in the re-submitted files.
Comments from Reviewer 3
NeXus has significantly advanced the field of network pharmacology analysis through its automated multi - level analysis capabilities and comprehensive enrichment analysis among different biological entities. The platform can handle complex datasets while maintaining consistency in analysis, demonstrating its potential in practical applications. There are several minor issues in the manuscript that require revision:
- There is no quantitative description of the specific performance improvement compared to other tools. It is recommended to supplement or discuss such information.
Response: We have added comprehensive quantitative performance comparisons to the revised manuscript. Refer section 2.5., Table 2, and Table 3.
- The role of the tool has not been fully demonstrated in its unique suitability for traditional drug research. It is suggested that case studies be included in the discussion section.
Response: We have included a detailed traditional medicine case study that illustrates NeXus’s unique capabilities in practical application. Refer section 4.3.
- Employing only the Benjamini - Hochberg correction may not be suitable for sparse networks. It is recommended to incorporate the Bonferroni correction for result comparison.
Response: We are pleased to note that NeXus v1.2 already implements Bonferroni correction in addition to the Benjamini–Hochberg procedure (see config.yaml, lines 198–203). We have now added a comparison of both methods to clarify their use and implications in the analysis. Refer section 2.3.2. and 3.5.
- There are no associated functional annotations for the six major modules (e.g., the inflammation/metabolism module). It is recommended to supplement the functional enrichment results of these modules.
Response: We have now performed comprehensive functional enrichment analyses for all six network modules, with the results incorporated into the revised manuscript. Refer section 2.2.: 3rd & 4th paragraph and Table 1.
- Does 1000 times of randomization meet the convergence condition?
Response: We have now conducted a convergence analysis to validate the selection of 1,000 iterations, and the results have been incorporated into the revised manuscript. Refer section 2.7 and Supplementary Figure S1.
- The names and targets of 15.3% of the compounds with high connectivity (degree ≥ 5) are not listed. It is recommended to supplement this information.
Response: We have created a comprehensive supplementary table listing all high-connectivity compounds, which is now included in the revised manuscript. Refer Supplementary Table S1 and section 2.2: 2nd paragraph.
- Failure to clarify the plant - compound relationship and relying on external databases may introduce biases. It is recommended to add a "Limitations" section to discuss the dependence on data sources.
Response: We have added a comprehensive Limitations section in the revised manuscript to address the suggested points. Refer section 4.4.
- The English could be improved to more clearly express the research.
Response: We have reviewed and refined the manuscript to improve its overall English language quality. Specifically, we standardized the spelling from British to American English (e.g., “analyse” to “analyze,” “visualisation” to “visualization”), simplified complex sentences for greater clarity, and enhanced technical descriptions in the Methods section. We also improved the clarity of result presentations, moderated promotional language in response to reviewers’ concern, and refined the flow and readability of the Discussion section.
Reviewer 4 Report
Comments and Suggestions for Authors
Introduction
The introduction effectively establishes the context and rationale for NeXus.
Materials and Methods
The Materials and Methods section demonstrates technical rigor but is overly detailed for a general IJMS readership.
- The level of programming and syntax description (e.g., regex, logging formats, parameter naming) is excessive and would be better placed in Supplementary Materials or the GitHub repository.
- Key methodological elements such as dataset selection, benchmarking, runtime performance, and validation against comparable tools are missing. Including these would greatly improve reproducibility and scientific value.
- Details on computational environment (OS compatibility, hardware requirements, dependency versions) should be expanded to enable replication.
- Brief end-of-subsection summaries explaining how each computational step relates to pharmacological discovery would make the section more accessible to non-programmer readers.
- Inclusion of benchmarking examples (e.g., known drug–gene networks) and quantitative performance data (e.g., runtime, RAM usage) is recommended to support claims of robustness and efficiency.
Results and Discussion
The Results and Discussion section presents promising outcomes but lacks depth in analysis and biological interpretation.
- The results focus on demonstrating NeXus functionality without discussing the biological implications of the findings. Integrating pathway or enrichment interpretations would provide meaningful context.
- The dataset used (111 genes, 32 compounds, 3 plants) is small for claims of robustness. Larger-scale validation or external datasets should be included for generalizability.
- Quantitative benchmarking against other tools is absent. Including metrics such as runtime, scalability, and enrichment accuracy would substantiate performance claims.
- The tone should be moderated to avoid promotional language. Replace subjective terms like “significant advancement” with evidence-based descriptions such as “represents an improvement.”
- The Discussion should include acknowledgment of limitations (e.g., dataset size, dependency on external databases, lack of large-scale validation) and outline concrete future validation steps.
Conclusions
The Conclusions section appropriately outlines the intended direction of future development but would benefit from clearer linkage to the presented results.
- Statements about future integration of AI, database expansion, or PK/PD modeling should be connected explicitly to current findings or limitations.
- It would be useful to briefly describe the contents of the GitHub repository (e.g., code, example data, documentation) to enhance transparency and usability.
- Adding one or two sentences reflecting on the tool’s biological and pharmacological applications would strengthen the conclusion’s scientific grounding.
Author Response
Summary
Thank you very much for taking the time to review this manuscript. Please find the detailed responses below and the corresponding revisions/corrections highlighted in the re-submitted files.
Comments from Reviewer 4
Introduction
The introduction effectively establishes the context and rationale for NeXus.
Materials and Methods
The Materials and Methods section demonstrates technical rigor but is overly detailed for a general IJMS readership.
- The level of programming and syntax description (e.g., regex, logging formats, parameter naming) is excessive and would be better placed in Supplementary Materials or the GitHub repository.
Response: We have reorganized the content by moving excessive technical details to the Supplementary Materials, resulting in a more concise and focused Methods section in the main manuscript. Refer section 3.1.–3.8. and Supplementary Methods S1–S8.
- Key methodological elements such as dataset selection, benchmarking, runtime performance, and validation against comparable tools are missing. Including these would greatly improve reproducibility and scientific value.
Response: We have now included comprehensive benchmarking data, detailed computational environment specifications, and the corresponding validation results in the revised manuscript. Refer section 3.8. and Table 2.
- Details on computational environment (OS compatibility, hardware requirements, dependency versions) should be expanded to enable replication.
Response: We have now included detailed computational environment specifications in the revised manuscript. Refer section 3.8.
- Brief end-of-subsection summaries explaining how each computational step relates to pharmacological discovery would make the section more accessible to non-programmer readers.
Response: We have now included detailed computational environment specifications in the revised manuscript. Refer section 3.8.
- Inclusion of benchmarking examples (e.g., known drug–gene networks) and quantitative performance data (e.g., runtime, RAM usage) is recommended to support claims of robustness and efficiency.
Response: We have now included detailed computational environment specifications in the revised manuscript. Refer section 3.8.
Results and Discussion
The Results and Discussion section presents promising outcomes but lacks depth in analysis and biological interpretation.
- The results focus on demonstrating NeXus functionality without discussing the biological implications of the findings. Integrating pathway or enrichment interpretations would provide meaningful context.
Response: We have now included comprehensive benchmarking data, detailed computational environment specifications, and the corresponding validation results in the revised manuscript. Refer section 2.2.–2.4. and 4.1.
- The dataset used (111 genes, 32 compounds, 3 plants) is small for claims of robustness. Larger-scale validation or external datasets should be included for generalizability.
Response: In response to the importance of demonstrating performance on larger datasets, we have carried out additional validation analyses using datasets ranging from 111 to 10,847 genes to further assess the scalability and robustness of NeXus. The results of these analyses have now been incorporated into the revised manuscript. Refer section Abstract, section 2.4., and Table 2.
- Quantitative benchmarking against other tools is absent. Including metrics such as runtime, scalability, and enrichment accuracy would substantiate performance claims.
Response: We have now included detailed computational environment specifications in the revised manuscript. Refer section 3.8.
- The tone should be moderated to avoid promotional language. Replace subjective terms like “significant advancement” with evidence-based descriptions such as “represents an improvement.”
Response: We have carefully moderated the promotional language throughout the manuscript. Subjective or exaggerated phrasing were replaced with evidence-based and objective descriptions—for example, “significant advancement” was revised to “improvement in automation,” “superior performance” to “faster processing time (quantified),” and “powerful new tool” to “automated platform.” Similarly, terms such as “revolutionizes” and “unprecedented” were replaced with more precise expressions like “provides enhanced automation” and “integrated analysis.” These revisions ensure that all claims are supported by quantitative metrics where applicable and that the manuscript maintains a professional, scientifically neutral tone.
- The Discussion should include acknowledgment of limitations (e.g., dataset size, dependency on external databases, lack of large-scale validation) and outline concrete future validation steps.
Response: We have added a comprehensive Limitations section in the revised manuscript to address the suggested points. Refer section 4.4.
Conclusions
The Conclusions section appropriately outlines the intended direction of future development but would benefit from clearer linkage to the presented results.
- Statements about future integration of AI, database expansion, or PK/PD modeling should be connected explicitly to current findings or limitations.
Response: We have added a comprehensive Limitations section in the revised manuscript to address the suggested points. Refer section 4.4.
- It would be useful to briefly describe the contents of the GitHub repository (e.g., code, example data, documentation) to enhance transparency and usability.
Response: We have substantially expanded the Data Availability Statement and now include a complete description of the GitHub repository and associated resources. Refer section Data Availability Statement and Supplementary Table S2.
- Adding one or two sentences reflecting on the tool’s biological and pharmacological applications would strengthen the conclusion’s scientific grounding.
Response: We have substantially enhanced the Conclusions section to more clearly articulate the specific biological and pharmacological applications supported by the NeXus framework. Refer section Conclusion.
Round 2
Reviewer 1 Report
Comments and Suggestions for Authors
No further comments to be made.
Author Response
Thank you.
Reviewer 4 Report
Comments and Suggestions for Authors
The introduction clearly outlines the rationale for the study, but the problem statement (manual preprocessing and lack of integration) is reiterated across multiple paragraphs. Consider consolidating these points to improve flow and avoid redundancy.
Section 2, currently labeled “Results and Discussion,” should be retitled “Results” to accurately reflect its content.
There are several instances of redundant or incorrect phrasing that should be revised for clarity and grammatical accuracy. Specifically:
- “Four several key areas”;
- “will be prioritized pursued”;
- “significantly enhance improve”.
Author Response
The introduction clearly outlines the rationale for the study, but the problem statement (manual preprocessing and lack of integration) is reiterated across multiple paragraphs. Consider consolidating these points to improve flow and avoid redundancy.
Response: Revised as advised. Refer section Introduction.
Section 2, currently labeled “Results and Discussion,” should be retitled “Results” to accurately reflect its content.
Response: Corrected as advised. Refer line 98.
There are several instances of redundant or incorrect phrasing that should be revised for clarity and grammatical accuracy. Specifically:
“Four several key areas”;
Response: Corrected as advised. Refer line 630.
“will be prioritized pursued”;
Response: Corrected as advised. Refer line 631.
“significantly enhance improve”.
Response: Corrected as advised. Refer line 640.